# Nutraceutical Properties of Polyphenols against Liver Diseases

**DOI:** 10.3390/nu12113517

**Published:** 2020-11-15

**Authors:** Jorge Simón, María Casado-Andrés, Naroa Goikoetxea-Usandizaga, Marina Serrano-Maciá, María Luz Martínez-Chantar

**Affiliations:** 1Liver Disease Laboratory, Center for Cooperative Research in Biosciences (CIC bioGUNE), Basque Research and Technology Alliance (BRTA), Bizkaia Technology Park, Building 801A, 48160 Derio, Bizkaia, Spain; ngoikoetxea@cicbiogune.es (N.G.-U.); mserrano@cicbiogune.es (M.S.-M.); mlmartinez@cicbiogune.es (M.L.M.-C.); 2Centro de Investigación Biomédica en Red de Enfermedades Hepáticas y Digestivas (CIBERehd), 48160 Derio, Bizkaia, Spain; 3Cell Biology and Histology Department, University of the Basque Country (UPV/EHU), Barrio Sarriena, S/N, 48940 Leioa, Spain; mdcasado002@gmail.com

**Keywords:** polyphenols, liver, stilbenes, flavonoids, phenolic acids, lignans, curcuminoids, NAFLD, HCC, DILI, ALF, ALD

## Abstract

Current food tendencies, suboptimal dietary habits and a sedentary lifestyle are spreading metabolic disorders worldwide. Consequently, the prevalence of liver pathologies is increasing, as it is the main metabolic organ in the body. Chronic liver diseases, with non-alcoholic fatty liver disease (NAFLD) as the main cause, have an alarming prevalence of around 25% worldwide. Otherwise, the consumption of certain drugs leads to an acute liver failure (ALF), with drug-induced liver injury (DILI) as its main cause, or alcoholic liver disease (ALD). Although programs carried out by authorities are focused on improving dietary habits and lifestyle, the long-term compliance of the patient makes them difficult to follow. Thus, the supplementation with certain substances may represent a more easy-to-follow approach for patients. In this context, the consumption of polyphenol-rich food represents an attractive alternative as these compounds have been characterized to be effective in ameliorating liver pathologies. Despite of their structural diversity, certain similar characteristics allow to classify polyphenols in 5 groups: stilbenes, flavonoids, phenolic acids, lignans and curcuminoids. Herein, we have identified the most relevant compounds in each group and characterized their main sources. By this, authorities should encourage the consumption of polyphenol-rich products, as most of them are available in quotidian life, which might reduce the socioeconomical burden of liver diseases.

## 1. Introduction

Current food tendencies and suboptimal dietary habits, together with an unhealthy lifestyle, are leading to the development of metabolic pathologies and their spreading worldwide [1,2]. In this context, the prevalence of liver pathologies is increasing among population, as this organ is responsible for the metabolism of exogenous substances in the organism [3]. Chronic liver pathologies, one of the leading mortality causes in USA and Europe, have on nutritional imbalances and sedentary habits their main causative agent nowadays. Non-alcoholic fatty liver disease (NAFLD) has emerged as the most frequent form of chronic liver disease worldwide, with an estimated prevalence of around 25% of general population [4,5]. Indeed, such elevated prevalence is expected to even increase within next years due to the rising of comorbidities from metabolic syndrome (MetS), making NAFLD a global health problem [6,7]. The term NAFLD is used to define a group of hepatic disorders that go from a simple lipid accumulation in the hepatocyte (steatosis) to its progression into more severe stages as non-alcoholic steatohepatitis (NASH), characterized by lipid-derived inflammation, hepatocellular ballooning and fibrosis. In case of a chronic fibrosis development, hepatocyte cell death and extracellular matrix (ECM) deposition, NASH may turn into cirrhosis. Moreover, the risk of developing NAFLD highly rises up the risk of developing hepatocellular carcinoma (HCC), the most frequent form of liver cancer [6,8,9,10].

Until date, the two-hit or multiple-hit hypothesis is the most extended explanation for the progression of NAFLD, in which a first hit induced steatosis and the aberrant lipid homeostasis leads to derived complications that contribute to its aggravation [11]. Related to the first hit, two imbalances have been reported to promote hepatic lipid accumulation, between: (i) fatty acid uptake and very-low-density lipoprotein (VLDL) export and (ii) de novo lipogenesis and fatty acid oxidation (FAO). Indeed, the metabolic triggering of the pathology has led to propose a new term MAFLD, metabolic-associated fatty liver disease, to define this group of pathologies [12]. Then, the appearance of second hits such as peroxidation, oxidative and reticulum stress development and mitochondrial dysfunction triggers an inflammatory response that may result in fibrosis development. In this process, the hepatocyte suffers from an antioxidant machinery depletion that finally leads to its death and, in the meantime, macrophage activation by pro-inflammatory cytokines such as tumor-necrosis factor (TNF) or several interleukine (IL) isoforms. Thus, hepatic stellate cells (HSC) are activate and proliferate by several signaling pathways such as transforming growth factor-beta (TGF-β)/SMAD, promoting collagen synthesis and ECM deposition, in which the matrix metalloproteinases (MMP)/tissue inhibitor of metalloproteinases (TIMP) is essential [13]. Regarding HCC development, the heterogeneity of the disease implies different molecular signaling pathways activated at the same time to deregulate hepatocyte growth, proliferation, differentiation and apoptosis. Several pro-proliferative pathways and signaling occur such as protein kinase B (AKT), nuclear factor-kappa B (NF-κB), mammalian target of rapamycin (mTOR) or c-MYC [14].

Furthermore, unhealthy lifestyle does not necessarily mean an inadequate food intake, but also into the excessive consumption of certain prescription and non-prescription medications or toxic compounds. As a consequence, liver can suffer from an acute liver failure (ALF) with drug-induced liver injury (DILI) as its main cause [15,16,17]. DILI is estimated to affect 14 of 100,000 inhabitants worldwide and it presents a real challenge to gastroenterologists when diagnosing the pathology [18]. The liver is the organ responsible of the metabolism of exogenous compounds. Under overdose conditions, compounds such as acetaminophen or carbon tetrachloride are converted by cytochrome P450 2E1 (CYP2E1) into toxic compounds by the hepatocyte [19]. These toxic compounds deplete the anti-oxidant machinery of the cell, mainly composed by reduced glutathione (GSH), catalase (CAT) and superoxide dismutase (SOD). The direct impact they have over mitochondrial integrity causes a damage that finally results on the necrosis of the hepatocyte [20,21]. During DILI, the release of mitochondrial pro-apoptotic proteins such as BAX or BCL-2 and the TNF- or NF-κB-mediated pro-inflammatory signaling are key hallmarks [22].

Additionally, the chronic and heavy consumption of alcohol leads to the development of steatosis in 90% of patients who drink over 60 g of alcohol per day and cirrhosis in 30% cases [23], making alcoholic liver disease (ALD) to follow a similar pattern of progression as NAFLD. Similarly to DILI, CYP2E1-mediated metabolism of ethanol leads to the production of acetaldehyde that leads to mitochondrial dysfunction [24] that impairs lipid homeostasis in the hepatocyte causing steatosis. The increased oxidative stress and depletion of anti-oxidant activity of the hepatocyte, together with aberrant lipid metabolism by peroxidation, induce a hepatocellular damage that promotes the progression of the disease from alcoholic steatosis to hepatitis and finally cirrhosis [24]. The molecular basis of ALD progression from steatosis to cirrhosis follow similar molecular mechanisms to NAFLD, including an inflammatory environment and HSC proliferation and activation [24].

Considering the elevated prevalence of aforementioned liver pathologies and their expected increase, together with the lack of awareness of general population, authorities are focusing on reducing their prevalence and improving their prognosis [25]. Clinical and scientific studies point out lifestyle modifications as the mainstay and cornerstone in treating these pathologies, comprising adequate meal plans and physical activity [26,27]. Although behavioral interventions attempt to guarantee the adherence of the patients, in most of cases it is hard to achieve so they do not follow the designed plans.

Therefore, the supplementation with certain products may offer a more easy-to-adhere approach in order to prevent or improve liver pathologies. In this context, current evidence highlights the beneficial properties associated to polyphenols, a group of natural metabolites contained in plants that own a variety of beneficial effects for the liver and associated comorbidities. They play a role in the regulation of oxidative stress, the lipid metabolism, the development of insulin resistance, inflammation or body weight among others [28,29]. Moreover, they are capable of attenuate drug-induced toxicity by reducing apoptosis and enhancing the expression of antioxidant enzymes [30]. Thus, they offer an attractive nutraceutical approach not only for reducing the impact and prevalence of chronic liver diseases, but also for ameliorating the prognosis of acute liver alterations.

The aim of the present review is to highlight the benefits of polyphenols intake and identify the main polyphenol-rich sources. By this, we propose a change in dietary lifestyle pattern by presenting such polyphenol-rich foods, which can be easily introduced in the diet. Considering their nutraceutical value, they may represent a strategic approach in which future dietary guidelines and public health recommendations should be based on.

## 2. Polyphenols and Their Nutraceutical Value

Polyphenols are a large group of at least 10,000 different naturally occurring phytochemicals, with one or more aromatic rings and with one or more hydroxyl functional groups attached. They are secondary metabolites that represent a large and diverse group of substances abundantly present in vegetables, fruits, cereals, spices, teas, rizhomes, medical plants and flowers [29,31].

Although the diversity of their chemical structure makes their classification difficult, the number of phenol rings and the structural elements allows to distinguish between certain groups of polyphenols. So that, according to their structural similarities polyphenols can be grouped in stilbenes, flavonoids, phenolic acids, lignans and curcuminoids [31,32]. In the following work, the main polyphenolic compounds of each group, their beneficial properties for certain liver pathologies and their main food source will be deeply described.

### 2.1. Stilbenes

Stilbenes are phytochemicals, some of which are considered phytoalexins, mainly present in berries, grapes, peanuts and red wine. This group of polyphenols is composed by three main compounds: resveratrol and its derived compounds pterostilbene and piceatannol [32,33].

Resveratrol may be one of the most popular polyphenols in our society and it is found in coco, mulberries, peanuts, soy and grapes [34]. Preclinical studies have characterized its protective features at multiple levels, by modulating oxidative stress and hepatocellular damage in order to ameliorate NAFLD through the reduction of free radicals and pro-inflammatory cytokines and the increased response of anti-oxidant enzymes such as glutathione (GSH) and cytochrome P450 (CYP) 2E1 [35,36]. Moreover, resveratrol reduces hepatic lipid content by reducing sirtuin 1 (SIRT1)-mediated lipogenic activity through the modulation of acyl-coA carboxylase (ACC), peroxisome proliferation activity receptor γ (PPARγ) and sterol response element binding protein-1 (SREBP-1) [37].

As aforementioned, pterostilbene is a derivate from resveratrol which is mainly present in blueberries [38]. This compound is also reported to reduce steatosis and modify hepatic fatty acid profile stimulating carnitine-palmitoyltransferase-1 (CPT1)-mediated FAO, stimulating microsomal triglyceride transfer protein (MTP)-mediated very-low-density lipoprotein (VLDL) export and reducing lipid uptake by CD36 [39]. Likewise, it enhances liver glucokinase and glucose-6-phosphatase activity to ameliorate insulin resistance and hepatic glycogen homeostasis and, therefore, lowering total cholesterol and triglyceride levels in serum [40].

Another derivate from the hydroxylation of resveratrol is piceatannol, present in grapes, passion fruit and peanut calluses [33]. Although this compound has been less studied than resveratrol due to its lower concentration in food, it has been reported to have a higher activity [41]. Thus, piceatannol also improves hepatic glycemic control by activating adenosine monophosphate-activated protein kinase (AMPK) through phosphorylation while ameliorating serum lipid profile in mice inhibiting the lipogenic flux mediated by ACC and fatty acid synthase (FAS) expression [42] Piceatannol-mediated AMPK phosphorylation also induces autophagy, a process reported to be dysregulated in NAFLD [43].

Regarding the effects of stilbenes among human population, clinical trials have been carried out only by evaluating the properties of resveratrol in NAFLD, liver cancer and hepatitis patients. Remarkably, the dietary supplementation with resveratrol has been shown to be effective in improving the inflammatory marker profile in NAFLD patients [44].

### 2.2. Flavonoids

Flavonoids comprise the larger group of polyphenols and the most abundant compounds in human diet. They are characterized by a C6-C3-C6 backbone structure and appear in almost all foods of vegetable origin and, particularly, in apples, berries, citrus fruits, onions, red wine, grapes, tea or olive oil [31]. Flavonoids are classified into six additional subgroups: anthocyanins, flavanols, flavanones, flavonols, flavones and isoflanoids. In the following section a detailed description of each subgroup and their main compounds is provided.

First, the subgroup of anthocyanins is composed by water-soluble flavonoid species as delphinidin, pelargonidin, cyanidin and malvidin. Delphinidin appears in flowers and berries as blueberry, Saskatoon berry, raspberry, strawberry or chokecherry, being its richest natural source the Maqui berry [45]. They have been reported to have anti-inflammatory properties targeting nuclear factor kappa-B (NF-κB), activator protein-1 (AP-1) and cyclooxygenase-2 (COX-2) [46]. Moreover, delphindin prevents triglyceride accumulation in in vitro NASH models modulating AMPK and FAS [47] or to downregulate fibrogenic stimuli to prevent fibrosis development in preclinical models [48]. Therein, fibrogenic response is attenuated by a decreased oxidative stress development, increasing matrix metalloproteinase (MMP)-9 and metallothionein (MT) I/II expression [48]. Although pelargonidins have been less studied, their protective properties against lipopolysaccharide (LPS)-induced liver injury have been characterized by modulating the inflammatory pathway mediated by toll-like receptor (TLR) [49]. This polyphenolic compound is mainly present in orange- or red-color fruits as raspberries, blackberries, strawberries or plums [50]. On another hand, cyanidin have been reported to promote lipid oxidative flux by increasing CPT1 and PPARα expression to enhance FAO and by decreasing FAS and SREBP-1 expression to downregulate lipogenesis [51]. Cyanidin prevents fibrosis development inhibiting collagen type I synthesis and downregulating extracellular-regulated kinase 1/2 (ERK1/2) [52], while promotes cAMP-mediated protein kinase A (PKA) activation to induce glutathione (GSH) synthesis and protect the hepatocyte [53]. Additionally, hepatocellular damage derived from alcoholic toxicity is also prevented by activating AMPK, that induces autophagy [54]. Cyanidins are present in red berries, grapes, bilberry, blackberry, blueberry, cherry, cranberry, elderberry, hawthorn, loganberry, açaai berry and raspberry [55]. Similar to cyanidin, malvidin is present in red grapes, cranberries, blueberries and black rice. They have been reported to increase FAO in the same way as cyanidins [51], and, remarkably, to attenuate tumor growth in HCC by regulating BAX and caspase-3 for apoptosis; several cyclin isoforms and phosphatase and tensin homolog (PTEN) for proliferation and metastasis derived from MMP-2/9 activity [56].

Secondly, flavanols share a general chemical structure of two rings linked by three carbons forming an oxygenated heterocyclic ring [57]. Among them epicatechin, epigallocatechin and its gallate derivate (EGCG) and procyanidins are the most popular compounds. Epicatechin is mainly present in dark chocolate and cocoa [58] and it has been reported to regulate lipid profile in serum and liver through regulating SREBP, FAS, liver X receptor (LXR) and SIRT [59]; as well as to attenuate oxidative stress and inflammatory injury via abrogation of NF-κB signaling pathway [60]. EGCGs, mainly present in green tea [61], may be another one of the most popular polyphenols in society normally sold as green tea extract. Their biological effects on NAFLD have been characterized in terms of lipid metabolism via pAMPK, SREBP-1, FAS and ACC; the oxidative response mediated by CYP2E1 or malonaldehyde production; TNF and IL-mediated inflammation and the fibrosis development induced by TGF-β/SMAD pathway [62]. EGCG also decreases body weight and reduces liver injury mediated by oxidative stress and inflammatory response, reducing the formation of collagen and alpha-smooth muscle actin (αSMA) in the liver and the expression of tissue inhibitor of metalloproteinase-2 (TIMP-2) in preclinical studies [63]. Moreover, EGCG has a protective effect on hepatotoxicity by decreasing bile acid and lipid absorption [60] and lowering cytochrome P450 (CYP)-mediated activation and toxicity of acetaminophen in DILI [64]. Related to HCC, EGCG has been also characterized to promote apoptosis in cancer cells in a multifactor way targeting genes involved in initiation (like NF-κB or BCL-2), proliferation (like cMyc, ERK1/2 or DDR mechanisms) and invasion (like MMPs or COX-2). [65]. The antioxidant properties of the last compound, procyanidins, have been also reported in fibrosis animal models via inhibition of CYP2E1-mediated metabolism of toxic compounds and improving antioxidant capacity through GSH or superoxide dismutase (SOD) [66]. Additionally, procyanidins exert a protective effect against ALD ameliorating SREBP-1-mediated steatosis and inflammation via IL-6 or TNF [67], with a possible involvement in preventing mitochondrial dysfunction and apoptosis [68]. Procyanidins are present in chocolate, apples, red grapes and cranberries [69].

The subgroup of flavanones is smaller than the previous one, as only hesperidin and naringenin compose it. Both compounds are characterized by a double bond between C2 and C3 and the lack of the oxygenation in C3 [70]. On one hand, hesperidin is mainly found in citrus fruits (grapefruit, lemon, lime or orange) and peppermint [71,72]. Similarly to other flavonoids, this compound has been found to protect against fibrosis enhancing GSH and decreasing catalase (CAT) and SOD levels [73]. Likewise, hesperidin reduced development of hepatic oxidative stress, dyslipidemia and histological changes via decreasing lipid peroxidation and recovering hepatocyte antioxidant properties [74]. On the other hand, naringenin is mainly found in Mexican oregano [75]. This flavanone’s beneficial effects have been studied over DILI by downregulating caspase-3, BAX and BCL [76]. Hepatoxocity-induced fibrosis is also inhibited by naringenin, that inhibits the development of oxidative stress, the activation of HSC mediated TGF-β and the synthesis of ECM [77].

Flavonols present a large group of polyphenols in which quercetin is one of the most important flavonoids and, in addition, kaempferol, myricetin, isorhamnetin and galangin also compose this group. Quercetin is found in a variety of food that includes apples, berries, brassica vegetables, capers, grapes, onions, shallots, tea, tomatoes, many seeds and nuts [78,79]. This flavonol has been characterized to ameliorate fibrosis development by targeting NF-κB-mediated signal transduction, downregulating TNF, IL-6, IL-1β and IL-8 cytokines production [78], together with an increase of the antoxidant mechanisms mediated by GSH and IL-10 and decreasing lipid peroxidation in ALD [79]. Kampferol, present in tea, broccoli, apples, strawberries and beans [80] prevents tumor development by enhancing PTEN expression and inactivate PI3K/Akt/mTOR signaling in order to inhibit migration, proliferation and invasion [81]. Otherwise, CYP2E1 inhibition by kaempferol protects the hepatocyte against ALD development [82], whereas fibrosis development is attenuated by the inhibition of SMAD2/3 via the direct interaction between kaempferol and ATP-binding pocker of activing receptor-like kinase 5 (ALK5) [83]. Myricetin is found in berries, honey, vegetables, teas and wines [84]. This flavonolic compound has a regressive effect on steatosis development in preclinical NASH models by promoting NRF2-mediated mitochondrial functionality, which increases antioxidative enzyme activities and PPAR-mediated fat decomposition [85]. Miricetin-mediated YAP downregulation also leads this polyphenol to exert anti-tumoral properties [86]. Isorhamnetin also alleviates steatosis decreasing FAS activity and fibrosis development via TGF-β-mediated HSC activation and proliferation [87], while decreasing the production of lipoperoxide compounds in serum and liver [88]. This compound is present in pears, onion, olive oil, grapes, tomato and the spice, Mexican Tarragon [80,89]. The last flavonol, galangin, is less abundant in nature as it is mainly present in galangal rizhome and propolis [90]. Similar to myricetin, galangin-mediated NRF2 activation attenuates oxidative damage, inflammation and apoptosis during hepatoxicity [91], while inhibiting the proliferation of HCC cells through the combined activation of NRF2 and hemooxygenase-1 (HO-1) [92].

The fifth flavonoid subgroup are flavones, distinguished by their double bond between C2 and C3, the lack of substitution at the C3 and the oxidation in C4 [93]. In this subgroup apigenin, chrysin and luteolin are the most relevant compounds. Apigenin is present in vegetables as parsley, broccoli, celery and onions; in fruits as oranges, olives, cherries and tomatoes; in herbs as chamomile, thyme, oregano, basil; and plant-based beverages as tea [93]. Between the beneficial properties of apigenin, it should be noted its anti-inflammatory properties against ALD by regulating CYP2E1-mediated oxidative stress and PPARα-mediated lipogenic gene expression [94] and the prospective effect for the damage induced by ischemia-reperfusion by suppressing inflammation, oxidative stress and apoptosis mediated by BAX and BCL-2 [95]. Additionally, this compound has been also characterized to ameliorate serum and hepatic lipid profile via metabolic and transcriptional modulations in the liver in genes involves in FAO, tricarboxylic acid cycle and oxidative phosphorylation among other [96]. Chrysin is specially present in honey and propolis [97] and this flavone has been reported to ameliorate NAFLD by modulating TNF- and IL-6-derived inflammatory response and SREBP-1-mediated lipogenesis in rats [98] and to reduce fibrosis development in a dose-dependent way via regulating MMP/TIMP imbalance [99]. Otherwise, luteolin is found in vegetables and fruits such as celery, parsley, broccoli, onion, carrots, peppers, cabbages or apple skins [100]. The protective properties of luteolin have been studied in DILI, where it restores the synthesis of antioxidant compounds as GSH while decreasing the inflammation signaling via TNF, NF-κB and IL-6 signaling and decreasing endoplasmic reticulum stress as well [101]. It also protects from developing liver pathologies derived from the chronic consumption of toxic substances as mercury, promoting mitochondrial functionality via NRF-2/NF-κB/P53 signaling [102] or alcoholic liver disease (ALD), where it downregulates the expression of SREBP-1 and recovers the AMPK activity [103].

The last subclass of flavonoids are isoflavonoids, where genistein and daidzein are the most common compounds. Genistein is found in soybeans and soy-based food and formulas, nuts and legumes as peas or lentils [104]. Its protective properties have been characterized on NAFLD by modulating PPARα-mediated lipid metabolism [105], while it also ameliorates hepatic inflammation by reducing TLR4 expression [106] and fibrosis development by decreasing lipid peroxidation and increasing GSH levels [107]. Similarly to genistein, daidzein is also found in the same food sources and the supplementation of daidzein, although it is less effective [108], has been reported to alleviate NAFLD by upregulating FAO and downregulating TNF expression [109].

Regarding the clinical trials carried out to determine the effect of flavonoids in human population, the effect of hesperidin supplementation has been studied in NASH development finding an improvement in steatosis, hepatic enzymes and several parameters as glycaemia [110]. A clinical study about naringenin has proposed this compound as an attractive approach for treating hepatitis C [111], while quercetin has been characterized to attenuate the secretion of the virus [112]. Additional clinical studies expected within next years will evaluate the effect of camu, a food rich in procyanidins, in obesity-related disorders as NAFLD and the effect of EGCG in cancer development from cirrhosis.

### 2.3. Phenolic Acids

This group of polyphenols is constituted by phenolic compounds, having one carboxylic group and typically in bound form as amides, esters or glycosides. They are found in a variety of plant-based foods, seeds, skins or fruits and leaves of vegetables [113]. In the meantime, phenolic acids are divided into hydroxibenzoic acids, hydroxycinnamic acids and oleuropeunosides.

On one hand, hydroxibenzoic acids possess a common structure of C6-C1 derived from benzoic acid [113], being ellagic and gallic the most common compounds. Ellagic acid may be the most common compound in this subclass and it is present in nuts, walnuts, berries and fruits as pomegranates or berries [114]. This molecule has been reported to normalize the activity of antioxidative enzymes and to ameliorate histopathology by reducing inflammatory response via modulating oxidative stress [115], also reducing oxidative stress after ischemia-reperfusion liver injury [116] or impeding hepatotoxicity-derived fibrosis development in preclinical studies via downregulating caspase-3, BCL-2 and NF-κB expression while elevating NRF-2-mediated mitochondrial functionality [117]. Similarly to ellagic acid, gallic acid is found in berries as blueberries and strawberries, and fruits as mango [118]. This compound has been reported to exert protective properties in liver damage induced by drug abuse by reducing TNF-mediated inflammation and lipid peroxidation [119]. Moreover, gallic acid increases GSH and CAT antioxidative activities to protect the hepatocyte from ischemia-reperfusion [120] and decreases fibrosis development by restoring GSH and TGF-β levels while normalizing HSC activation and proliferation [121].

On the other hand, hydroxycinnamic acids derive from cinnamic acid and they are often present in food as simple esters with quinic acide or glucose [113], being ferulic and chlorogenic acids the most frequent compounds. Ferulic acid is found in commelinid plants as rice, wheat, oats or grains, and in vegetables, pineapple, beans, coffee, artichoke, peanut or nuts [122]. Similarly to hydroxybenzoic compounds, it upregulates NRF-2/HO-1 signaling to restore mitochondrial integrity and reduce the development of oxidative stress and inflammation in DILI [123], whereas it prevents fibrosis development by interfering in TGF-β/SMAD-mediated activation of HSCs [124]. Chlorogenic acid is particularly found in the coffee grain but it is also present in beans, potato tubers, fruits as apple and prunes [125]. This hydroxycinnamic compound also fibrosis development mediated by pro-inflammatory citokines such as TNF, IL-6 and IL-1β [126] and scavenges ROS production in alcohol consumption, reducing the steatosis, apoptosis and fibrosis development pathways mediated by TNF and TGF-β [127].

Oleuropein is mainly present in olive leaves, olives, virgin olive oil and olive mill waste [128]. Interestingly, this polyphenol has been shown to exert anti-inflammatory properties by scavenging ROS production under hepatotoxic conditions [129] and reduce lipid-derived inflammatory processes to prevent NASH progression such as TLR-mediated response [130].

Concerning the properties of phenolic acids in the human organism, clinical trials have been only developed by evaluating NAFLD development with a gallic acid-rich compound (Ajwa Date) and coffee supplementation, rich in chlorogenic acid. Although liver diseases were studied in the clinical trial evaluating Ajwa Date, outcomes have been focused on the prevention of atherosclerosis development. The results from the other clinical trial with coffee supplementation have not been published yet.

### 2.4. Lignans

Lignans are characterized by two phenylpropane units linked by a C6-C3 bond between the central atoms of the respective side chains. This group of polyphenols is present in a wide variety of plans in which latter, flaxseed and sesame seed represent the richest sources [131]. Moreover, lignans can be also found in fish, whole-grain cereals (as wheat or oats), meat, oilseed (as flax or soy) and beverages (as coffee, tea or wine) [131]. Although it can be distinguished among classical lignans, neolignans, flavonolignans and carbohydrate-conjugates, the main compounds present in nature are sesamin and diglucoside.

Sesamin is mainly present in sesame seeds and preclinical studies have reported metabolic properties in liver pathologies by preventing from ACC- [132] and SREBP-1-mediated fatty acid synthesis [133], while enhances FAO mediated by CPT1 or 3-hydroxyacyl-coA dehydrogenase [132]. Otherwise, diglucoside is found in flaxseed [134] and this compound has been also reported to downregulate hepatic lipid accumulation, while downregulating hepatic lipid peroxidation and decreasing cholesterol in serum [135].

Until date, no clinical trials for evaluating lignans have been carried out.

### 2.5. Curcuminoids

Regarding the group of curcuminoids, curcumin is the main compound as it gives the name to this group. This compound is the principal extract from the turmeric (Curcumula longa) herb [136] and preclinical approaches have characterized its anti-inflammatory and anti-oxidant properties derived from the intake of hepatotoxic compounds [137]. Curcumin alleviates hepatic dyslipidemia by inhibiting lipogenesis and promoting FAO, while enhancing cholesterol efflux and, in the meantime, reducing the lipid imbalance-derived oxidative stress [137]. By this, the expression of NRF-2 restores mitochondrial integrity in the hepatocyte, while GSH increase leads to an enhanced antioxidant capacity thus downregulating HSC activation [137].

There are currently three clinical trials under recruitment in order to evaluate the effects of different forms of curcumin, as dietary supplement or conjugated to phosphatidylcholine, in the development of NAFLD and insulin resistance. Another clinical trial has proven its effectivity in reducing steatosis, reducing body-mass index and improving serum profile in terms of cholesterol, triglycerides and transaminases [138].

## 3. Discussion

It is a fact that current unhealthy food tendencies, accompanied by a more sedentary lifestyle, have a direct impact over the health of global population [1]. Metabolic disorders are spreading worldwide and, among them, liver pathologies are on the most extended ones. Non-alcoholic fatty liver disease or NAFLD has an alarming prevalence of 25% worldwide and it is even expected to increase within next years due to such unhealthy lifestyle [5]. Otherwise, the excessive drug consumption that sometimes takes place can also lead to other liver pathologies, reaching an acute liver failure (ALF) in which drug-induced liver injury (DILI) is the main cause affecting [18] and chronic alcohol consumption leads to the development of alcoholic liver diseases (ALD) [23]. The management of liver pathologies presents a challenge to authorities and, although dietary and behavioral plans are currently being carried out, the long-term compliance of population sometimes presents the true challenge. Therefore, the supplementation or feeding with certain products can offer a more easy-to-adhere strategy in terms of preventing or ameliorating both chronic and acute liver diseases. Related to this, in the present work the role of different polyphenols has been described in detail as well as the most relevant clinical trials about them (Table 1). Overall, all polyphenols [26,31] described in the present review are reported to have beneficial properties towards either preventing or ameliorating NAFLD, DILI or ALD. Although some of them as resveratrol, EGCG, or curcumin are more popular in society, any of these compounds may offer healthy properties for the liver.

Thus, the consumption of polyphenol-rich food is a suitable option when planning a diet. As it can be observed in Table 2, most of them are present in foods that can be easily found in any supermarket so general population might not have problems when acquiring them. Therefore, it is an interesting point that authorities promote the consumption of these kind of foods when designing their programs for creating awareness, especially in such patients of liver diseases who are under treatment. Reducing their prize or promoting their inclusion in certain products or meals (e.g., strawberries in yogurts or coffee in some drinks) might be adequate options. Moreover, as it can be observed in Table 2, most part of the polyphenol-rich foods are not high-calorie so their inclusion should not have an impact over total daily calorie intake, another concern in the development of MetS and related metabolic disorders [139]. Furthermore, it must be always taken into account that not only polyphenols but also other micronutrients present beneficial properties, and the existence of variety in a diet is which makes it healthy.

## 4. Conclusions

The supplementation with polyphenols has an effect in treating liver pathologies: non-alcoholic fatty liver disease, drug-induced liver injury, hepatocellular carcinoma and alcoholic liver disease. The inclusion of polyphenol-rich foods is an attractive approach when developing a nutritional program. Authorities should encourage their consumption. Polyphenols and other micronutrients are essential for an equilibrated diet, where variety is an essential feature.

## Figures and Tables

**Table 1 nutrients-12-03517-t001:** Clinical trials testing polyphenols against liver pathologies.

Polyphenol	Group/Subgroup	Pathology	Outcome
Resveratrol	Stilbenes	NAFLD, HCC, Hepatitis	Improved inflammatory profile in NAFLD [44].
Hesperidin	Flavonoids/Flavanones	NASH	Ameliorated steatosis, hepatic enzymes and glycaemia [110].
Naringenin	Flavonoids/Flavanones	Hepatitis C	Ameliorated phenotype [111].
Quercetin	Flavonoids/Flavonols	Hepatitis C	Attenuated secretion of the virus [112]
Procyanidins	Procyanidins/Flavanols	NAFLD	Not finished
EGCG	Flavonoids/Flavanols	Cirrhosis-derived HCC	Not finished
Gallic acid	Phenolic acids/Hydroxibenzoic acids	NAFLD	Atherosclerosis reduction.
Chlorogenic acid	Phenolic acids/Hydroxicinnamic acids	NAFLD	Not published
Curcumin	Curcuminoids	NAFLD	Reduction in steatosis and body-mass index and improved serum profile [138]

**Table 2 nutrients-12-03517-t002:** List of most relevant polyphenols, their richest sources and pathologies with potential beneficial properties with each respective molecular target.

Polyphenol	Group/Subgroup	Source	Liver Pathology	Molecular Targets
Resveratrol	Stilbenes	Coco, mulberries, peanuts, soy and grapes [34]	Steatosis/NASH	Glutathione, CYP2E1 [35,36]
			Steatosis	SIRT1, ACC, PPARγ, SREBP-1 [37]
Pterostilbene	Stilbenes	Blueberries [38]	Steatosis	Glucokinase, Glucose-6-phosphatase [40]
			Steatosis	CPT1, MTP, CD36 [39]
Piceatannol	Stilbenes	Grapes, passion fruit and peanut calluses [33]	Steatosis	AMPK, ACC, FAS and autophagy [42]
Delphinidin	Flavonoids/anthocyanins	Flowers, blueberry, Saskatoon berry, raspberry, strawberry, chokecherry, Maqui berry [45]	NASH/ALD	NF-κB, AP-1, COX-2 [46]
			Steatosis	AMPK, FAS [47]
			Fibrosis	Oxidative stress, MMP-9 and MT [48]
Pelargonidin	Flavonoids/Anthocyanins	Raspberries, blackberries, strawberries or plums [50]	NASH/ALD	TLR [49]
Cyanidin	Flavonoids/Anthocyanins	Red berries, grapes, bilberry, blackberry, blueberry, cherry, cranberry, elderberry, hawthorn, loganberry, açaai berry and raspberry [55]	Steatosis	CPT1, PPARα, FAS, SREBP-1 [51]
			Fibrosis	Collagen I, ERK 1/2 [52]
			NASH/Fibrosis	PKA, GSH [53]
			ALD	AMPK [54]
Malvidin	Flavonoids/Anthocyanins	Red grapes, cranberries, blueberries and black rice [80]	Steatosis	CPT1, PPARα, FAS, SREBP-1 [51]
			HCC	BAX, Caspase-3, Cyclin, PTEN, MMP-2/9 [56]
Epicatechin	Flavonoids/Flavanols	Dark chocolate and cocoa [58]	Steatosis	SREBP-1, FAS, LXR, SIRT [59]
			DILI/ALD	Bile acid and lipid absorption [60]
Epigallocatechin/EGCG	Flavonoids/Flavanols	Green tea [61]	NASH	NF-κB [60]
			Steatosis/NASH	AMPK, SREBP-1, FAS, ACC; CYP2E1, malonaldehyde, TNF, IL; TGF/SMAD [62]
			Fibrosis	Collagen, αSMA, TIMP-2 [63]
			DILI	CYP [64]
			HCC	NF-κB, BCL2; cMYC, ERK1/2, DDR; MMP, COX-2 [65]
Procyanidins	Flavonoids/Flavanols	Chocolate, apples, red grapes and cranberries [69]	NASH/Fibrosis	CYP2E1. GSH, SOD [66]
			ALD	SREBP-1, IL-6, TNF [67]
			NASH/ALD/DILI	Mitochondrial dysfunction and apoptosis [68]
Hesperidin	Flavonoids/Flavanones	Citrus fruits and peppermint [71,72]	NASH/Fibrosis	GSH, CAT, SOD [73]
			Steatosis/NASH	Lipoperoxidation [74]
Naringenin	Flavonoids/Flavanones	Mexican oregano [75]	DILI	Caspase-3, BAX, BCL [76]
			Fibrosis	TGF-β, ECM deposition [77]
Quercetin	Flavonoids/Flavonols	Apples, berries, brassica vegetables, capers, grapes, onions, shallots, tea, tomatoes, seeds and nuts [78,79]	Fibrosis	NF-κB, TNF, IL-1β, IL-6, IL-8 [78]
			ALD	GSH, IL-10, lipid peroxidation [79]
Kaempferol	Flavonoids/Flavonols	Tea, broccoli, apples, strawberries and beans [80]	Fibrosis	ALK5, SMAD 2/3
			HCC	PTEN, PI3K/AKT/mTOR [81]
			ALD	CYP2E1 [82]
Myricetin	Flavonoids/Flavonols	Berries, honey, vegetables, teas and wines [84]	Steatosis/NASH	NRF-2, mitochondrial functionality, PPAR [85]
			HCC	YAP [86]
Isorhamnetin	Flavonoids/Flavonols	Pears, onion, olive oil, grapes, tomato, Mexican Tarragon [80,89]	Steatosis/NASH/Fibrosis	FAS, TGF.β, HSC activation [87]
			NASH	Lipoperoxidation [88]
Galangin	Flavonoids/Flavonols	Rizhome and propolis [90]	NASH/DILI	NRF-2, apoptosis [91]
			HCC	NRF-2, HO-1 [92]
Apigenin	Flavonoids/Flavones	Parsley, broccoli, celery, onions, oranges, olives, cherries, tomatoes, chamomile, thyme, oregano, basil, tea [93]	ALD	CYP2E1, PPARα [94]
			Steatosis	FAO, Tricarboxylic acid cycle, oxidative phosphorylation [96]
Chrysin	Flavonoids/Flavones	Honey and propolis [97]	Steatosis/NASH	TNF, IL-6, SREBP-1 [98]
			Fibrosis	MMP, TIMP [99]
Luteolin	Flavonoids/Flavones	Celery, parsley, broccoli, onion, carrots, peppers, cabbages and apple [100]	DILI	GSH, TNF, NF-κB, IL-6, ER stress [101]
			Fibrosis/DILI	NRF-2, NF-κB, P53 [102]
			ALD	SREBP-1, AMPK [103]
Genistein	Flavonoids/Isoflanoids	Soybeans, nuts and legumes [104]	Steatosis	PPARα [105]
			NASH	TLR4 [106]
			Fibrosis	Lipoperoxidation, GSH [107]
Daidzein	Flavonoids/Isoflanoids	Soybeans, nuts and legumes [104]	Steatosis/NASH	FAO, TNF [109]
Ellagic acid	Phenolic acids/Hydroxibenzoic acids	Nuts, walnuts, berries, pomegranades or berries [114]	NASH/DILI/ALD	Oxidative stress [115]
			IR	Oxidative stress [116]
			Fibrosis	Caspase-3, BCL-2, NF-kB, NRF-2 [117] aslan
Gallic acid	Phenolic acids/Hydroxibenzoic acids	Blueberries, strawberries and mango [118]	Fibrosis	GSH, TGF-β [121]
			DILI/ALD	TNF, lipoperoxidation [119]
			IR	GSH and CAT [120]
Ferulic acid	Phenolic acids/Hydroxycinnamic acids	Rice, wheat, oats, grains, vegetables, pineapple, beans, coffee, artichoke, peanut, nuts [122]	DILI	NRF-2/HO-1 [123]
			Fibrosis	TGF-β/SMAD [124]
Cholorogenic acid	Phenolic acids/Hydroxycinnamic acids	Coffee, beans, potato, apple and prunes [125]	Fibrosis	TNF, IL-6 and IL-1β [126]
			ALD	ROS, TNF, TGF-β [127]
Oleuropein	Phenolc acids/Oleuropeunosides	Olive leaves, olives, virgin olive oil and olive mill waste [128]	DILI/ALD	ROS [129]
			NASH	TLR [130]
Sesamin	Lignans	Flaxseed and sesame seeds [131]	Steatosis	ACC, CPT1, 3-hydroxyacyl-coA dehydrogenase [132]
			Steatosis	SREBP-1 [133]
Diglucoside	Lignans	Flaxseed [134]	Steatosis/NASH	Lipoperoxidation [135]
Curcumin	Curcuminoids	Curcuma longa [136]	Steatosis	FAO [137]
			Fibrosis/DILI/ALD	NRF-2, GSH, HSC activation [137]

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
