# Peer review of "Nutraceutical Properties of Polyphenols against Liver Diseases"

_nutrients, 2020, doi:10.3390/nu12113517_

Round 1
Reviewer 1 Report
Overall, this is a clear and well-written review. The authors make an overview of many polyphenols of interest for human health with particular interest on the beneficial effects on damage caused by liver diseases, and they suggest a healthy diet, rich in these natural compounds.
However, this reviewer believes that some topics require to be revised.
Minor revision:
- The authors could cite works concerning another polyphenol with beneficial effects in liver diseases, oleuropein, a polyphenol extracted from the leaves and fruit of the olive tree.
- The authors describe Non-alcoholic fatty liver disease (NAFLD) but do not mention that a new nomenclature for this disease has been proposed and that better describes the pathology: Metabolic Associated Fatty Liver Disease (MAFLD) (Eslam M, Sanyal AJ, George J; International Consensus Panel. MAFLD: A Consensus-Driven Proposed Nomenclature for Metabolic Associated Fatty Liver Disease. Gastroenterology. 2020 May;158(7):1999-2014.e1. doi: 10.1053/j.gastro.2019.11.312. Epub 2020 Feb 8. PMID: 32044314. This reviewer suggests mentioning this new nomenclature as well.
Author Response
We thank the reviewer for considering our work suitable for publication and his/her constructing comments provided.
We have included the polyphenol oleuropein suggested. In agreement with the reviewer, it is present in several foods from our diet so it is of relevance when mentioning important phenolic compoinds.
Regarding the term MAFLD, although this term is beginning to be adopter by several experts in hepatology to define the spectrum of diseases with a metabolic origin, we think that the term NAFLD is more helpful to define them as it is more widely adopted. Moreover, NAFLD also allows to distinguish better from ALD, frequently mentioned during this work.
Reviewer 2 Report
Comments on Simon et al.
This manuscript covers the important aspects of polyphenols for liver diseases. However, there are still some issues that need to be addressed.
Major comments:
The major part of this manuscript covers the therapeutic effects of polyphenols but relatively small discussion on liver diseases. As a reader, I expect more detail information of liver diseases before going to the polyphenol part. My suggestion is to define liver diseases in detail at the beginning. Please discuss the molecular mechanisms (inflammation, oxidative stress, fibrosis, etc.) of common types of liver diseases that may facilitate the molecular targets of polyphenols. In this section, please provide a diagram of molecular mechanisms of liver diseases.
In the polyphenol parts, please provide the main mechanisms of action of polyphenol against particular liver diseases. Please avoid to say general statement like polyphenol reduces inflammation. Please explain how polyphenol reduces inflammation (maybe by suppression of IL-6 or IL-11, or other cytokines and chemokines). It should be applied for other pathological processes, such as oxidative stress, fibrosis, etc.
Authors are encouraged to provide a summary figure with key mechanisms of liver diseases that are targeted by polyphenols.
Table 1 needs to be reorganized. It needs an extra column for “molecular targets” or “therapeutic effects”. Please also add an extra column for experimental models (such as immortalized cell lines or animal models or clinical trials with phase and identification number).
Please revise these sentences “…………Consequently, the prevalence of liver pathologies is increasing, as it is the main metabolic organ in the body: chronic liver diseases. with non-alcoholic fatty liver disease (NAFLD) as the main cause, have an alarming prevalence of around 25% worldwide, whereas the consumption of certain drugs leads to an acute liver failure (ALF)”.
Author Response
We appreciate the reviewer's comments and we have worked deeply on them. Please find a point-by-point response bellow.
It is true that molecular targets and mechanisms had not been indicated in the previous version. In this new revised version we have included such molecular targets of polyphenols in different liver pathologies. Moreover, we have introduced the main molecular mechanisms by which NAFLD, DILI and ALD occur. As suggested, a diagram of molecular mechanisms in liver diseases has been also provided in the Graphical Abstract.
As suggested, a summary figure has been included as Graphical Abstract to make easier the reading of the manuscript.
Table 1 has been reorganized including the molecular targets of each polyphenol. We also have re-grouped each food by the respective polyphenol and indicated the group which it belongs to. We have considered that including an additional column with experimental models may be confusing for following the table. However, we have included clinical trials, which are the studies of the main relevante, in another Table 2.
The languaje has been exhaustively revised in order to avoid typos and grammar mistakes.
Reviewer 3 Report
The manuscript reviews the recent findings on the protective role of polyphenols in the development of liver diseases. The manuscript brings up an interesting topic to discuss, however, a couple of missing information need to be added:
The major comments are listed as follows:
- The biggest concern of this manuscript is missing the mechanismsthrough which polyphenols could potentially affect liver diseases. Please review the recent findings on the underlying molecular mechanisms of polyphenols in the treatment of liver diseases. It’s critical to provide readers with information on why polyphenols are potentially beneficial for the the liver disease patients.
- Please considering adding a diagram to show the general chemical structure of each type of polyphenols and their major dietary sources.
- Please considering adding a table to summarize the major findings of the clinical trials of polyphenolsin the treatment of liver diseases.
- To make it easier for the readers to follow, I would suggest to group the materials in Table 1 by the five types of polyphenols.
Some other minor issues are:
- Please make a meaningful title for Table 1.
- The section 2.4 is missing.
- There are multiple grammar mistakes and inappropriate expressions in the text. For example, Line 38-40, Line 63-65. Please considering asking a native English speaker to proofread the manuscript.
Author Response
We thank the reviewer for the constructing suggestions as they have allowed us to improve the quality of the manuscript. Please find herein a response of the main points:
1) The mechanisms through which polyphenols could affect liver diseases have been included. We also de have included them in a Table 1 in order to better summarize them.
2) Although the reviewer suggested to include the chemical structure of polyphenols, in the present manuscript we have tried to mainly focus on their action and their source. However, we have included a diagram related to the mechanisms of liver diseases and the actions of each polyphenol.
3) Following the reviewer's comment, a Table 2 has been included to highlight the clinical trials with polyphenols.
4) Table 1 has been reformulated including the molecular targets of each polyphenol. Additionally, polyphenols have been grouped in their respective group/subgroup, indicating the main source of each compound.
1) Table 1 has been given a meaningful tittle, as in the previous version it was not well indicated.
2) Sections have been reformulated to avoid the missing of a 2.4 section.
3) English revision has been performed by a native speaker in order to avoid such mistakes.
Round 2
Reviewer 3 Report
The authors have addressed all my questions properly. So I endorse the publication of this manuscript.